# High-Resolution Observations in the Western Mediterranean Sea: The REP14-MED Experiment

Reiner Onken[1], Heinz-Volker Fiekas[2], Laurent Beguery[3], Ines Borrione[4], Andreas Funk[2],
Michael Hemming[5,10], Jaime Hernandez-Lasheras[9], Karen J. Heywood[5], Jan Kaiser[5], Michaela Knoll[2],
Baptiste Mourre[9], Paolo Oddo[4], Pierre-Marie Poulain[6], Bastien Y. Queste[5], Aniello Russo[4],
Kiminori Shitashima[7], Martin Siderius[8], and Elizabeth Thorp Küsel[8]

[1]Helmholtz-Zentrum Geesthacht Centre for Materials und Coastal Research (HZG), Max-Planck-Straße 1, 21502 Geesthacht, Germany
[2]Wehrtechnische Dienststelle für Schiffe und Marinewaffen, Maritime Technologie und Forschung (WTD71), Berliner Straße 115, 24340 Eckernförde, Germany
[3]ALSEAMAR, 9 Europarc Sainte Victoire, 13590 Meyreuil, France
[4]Centre for Maritime Research and Experimentation (CMRE), Viale San Bartolomeo 400, 19126 La Spezia, Italy
[5]Centre for Ocean and Atmospheric Sciences, School of Environmental Sciences, University of East Anglia (UEA), Norwich Research Park, Norwich, NR4 7TJ, United Kingdom
[6]Istituto Nazionale di Oceanografia e di Geofisica Sperimentale (OGS), Borgo Grotta Gigante 42/c, 34010 Sgonico (Trieste), Italy
[7]Kyushu University, International Institute for Carbon-Neutral Energy Research (I2CNER), Fukuoka 819-0395, Japan
[8]Portland State University (PSU), PO Box 751, Portland, OR 97207-0751, United States of America
[9]Sistema de Observación Costero de las Islas Balears (SOCIB), Parc Bit, Naorte, 07121 Palma de Mallorca, Spain
[10]Laboratoire d'Océanographie et du Climat, 4, Place Jussieu, 75005 Paris, France.

*Correspondence to:* Reiner.Onken@hzg.de

**Abstract.** The observational part of the REP14-MED experiment was conducted in June 2014 in the Sardo-Balearic Sea west of Sardinia (Western Mediterranean Sea). Two research vessels collected high-resolution oceanographic data by means of hydrographic casts, towed systems, and underway measurements. In addition, a vast amount of data was provided by a fleet of 11 ocean gliders, time series were available from moored instruments, and information on Lagrangian flow patterns was obtained from surface drifters and 1 profiling float. The spatial resolution of the observations encompasses a spectrum over 4 orders of magnitude from $\mathcal{O}(10^1 \text{ m})$ to $\mathcal{O}(10^5 \text{ m})$, and the time series from the moored instruments cover a spectral range of 5 orders from $\mathcal{O}(10^1 \text{ s})$ to $\mathcal{O}(10^6 \text{ s})$. The objective of this article is to provide an overview of the huge data set which has been utilised by various studies, focusing on (i) water masses and circulation, (ii) operational forecasting, (iii) data assimilation, (iv) variability of the ocean, and (v) new payloads for gliders.

## 1 Introduction

The REP14-MED experiment was conducted in the framework of the *Environmental Knowledge and Operational Effectiveness* research programme of the Centre for Maritime Research and Experimentation (CMRE). REP14-MED was part of a series of experiments denoted by the acronym REP (*Recognized Environmental Picture*). The suffix 'MED' stands for *Mediterranean*, in

order to distinguish these experiments from another REP series conducted by the Portuguese Navy in the Atlantic. REP14-MED was led by CMRE, and the experiment was supported by 20 partners from 6 different NATO nations (Table 1). The activities at sea were conducted in June 2014 by the NATO Research Vessel *Alliance* and the Research Vessel *Planet* of the German Federal Ministry of Defence (Fig. 1), and comprised both oceanographic and acoustic observations. This article, however, describes

only the oceanographic measurements which are the basis of several research papers published in the special issue "REP14-MED: A Glider Fleet Experiment in a Limited Marine Area" of *Ocean Science.* In addition, standard names are introduced which may be referred to by other papers related to REP14-MED, e.g. for sections, tracks, and moorings. The overall goal of REP was to develop and test methods for the rapid characterisation of the marine environment, and the particular objectives of REP14-MED were to

1. Investigate water masses and circulation patterns off the west coast of Sardinia;

2. Collect data for operational ocean forecasting, model validation, and the evaluation of forecast skill;

3. Provide a data set for the development and comparison of different methods for data assimilation;

4. Employ data from a fleet of gliders for a cost/benefit analysis;

5. Conduct a generic experiment for the exploration of the variability of the ocean;

6. Test new payloads for gliders.

The selection of the experimental site was driven by the requirements of the acoustic experiments and the objectives defined above: the acoustic experiments required a wide shelf area, and the operations of the glider fleet should take place in an area with little ship traffic and without strong boundary currents. Therefore, the waters off the west coast of Sardinia (Western Mediterranean) were selected (Fig. 2). From a morphological point of view, this area is characterised by a wide continental

shelf area, the width of which varies between about 40 km and 80 km. The shelf ends at water depths between 150 m and 200 m, followed by the continental slope which features several canyons. The deep-sea area belongs to the Sardo-Balearic Basin and exhibits water depths of up to 2800 m.

The general circulation of the Western Mediterranean was comprehensively described by Millot (1999) and Robinson et al. (2001), and this picture was still valid at the beginning of the experiment. Accordingly, the mean surface circulation at the

experimental site is mainly related to the inflow of "new" Modified Atlantic Water (MAW) from the Strait of Gibraltar by means of anticyclonic eddies shed by the Algerian Current (Ribotti et al., 2004; Testor and Gascard, 2005; Escudier et al., 2016). Another branch of "old" MAW, which mixed with the underlying water masses on its large-scale cyclonic circulation through the Tyrrhenian, Ligurian, and Balearic Seas, comes probably from the west via the Balearic Current (García-Ladona et al., 1996). Just below the MAW, Winter Intermediate Water (WIW) follows the path of the MAW along its whole cyclonic

path. WIW is formed in late winter in the northern and northwestern Provençal Basin and it is supposed that it also finds a direct way from the formation sites to the Sardo-Balearic Basin via mesoscale eddies. Levantine Intermediate Water (LIW) originates from the Eastern Mediterranean and the direct path to the experimental site is via the Sicily Strait and the Sardinia Channel and then northward around the southern tip of Sardinia. After exiting the Strait of Sicily, another LIW branch joins the above mentioned circulation paths of "old" MAW. Below the LIW, Western Mediterranean Deep Water and Bottom Water are

found. Finally, the North Balearic Front (Testor and Gascard, 2003) represents the confluence zone between the waters coming

from the south and the waters from the north; according to Fuda et al. (2000) and Olita et al. (2013), it is located between about 40°N and 41°N.

The activities at sea took place within the red box indicated in Fig. 2 which is the so-called *observational domain*. In the west, it is bounded by the 7°15′E meridian, and in the east by the Sardinian coast. The southern and northern boundaries are at 39°12′N and 40°12′N, and the area size is approximately 60 nmi (nautical miles) x 60 nmi or 111 km x 111 km, respectively. A summary of the observations is given in Table 2.

## 2 Measuring systems

### Shipborne systems

Both *Alliance* and *Planet* are survey ships equipped with the standard oceanographic measuring systems. Only those systems which were used for the oceanographic experiments will be described (for more details, see Onken et al. (2014)). On *Alliance*, there were

– 2×*SBE* CTD (*Conductivity-Temperature-Depth*) probes with rosette sampler (for more details, see http://www.seabird.com)
– 1×*SBE* CTD probe with rosette sampler and optical sensors
– 1×*SATLANTIC* HyperPro optical profiler (http://satlantic.com)
– 2×towed ScanFish with dual CTD (http://www.eiva.com)
– 2×*RDI* ADCPs (*Acoustic Doppler Current Profiler*, http://www.rdinstruments.com), 300 kHz and 75 kHz
– 1×thermosalinograph (http://www.seabird.com)
– ship meteorological sensor systems

and on *Planet*

– 2×*SBE* CTD probe with rosette sampler
– 1×*OceanScience* underway CTD probe (*uCTD*, http://www.oceanscience.com)
– 1×towed CTD chain with 91 sensors 3.5 m apart (Sellschopp, 1997)
– 1×ADCP, 150 kHz
– ship meteorological sensor systems

### Gliders

16 gliders from different manufacturers were provided for launch during the experiment, of which 11 were used for the tasks described in this article. Another 3 were reserved for the acoustic experiments, and the remaining 2 served as backup:

– 9×*Slocum* (CMRE, http://www.webbresearch.com)
– 1×*Slocum* (PSU)
– 1×*Slocum* (WTD71)
– 3×*Seaglider* (UEA, http://www.km.kongsberg.com)
– 2×*SeaExplorer* (ALSEAMAR, http://www.alseamar-alcen.com)

For further details and the technical specifications, see Table 3.

**Surface drifters and profiling floats**

In total, 17 surface drifters and 2 profiling floats were available:

- 9×*Albatros* (http://www.albatrosmt.com) MD03i surface drifters (CMRE)
- 3×*Albatros* ODi-Solar surface drifters (CMRE)
- 5×SVP (Lumpkin, 2005) surface drifters (OGS)
- 2×ARVOR-I profiling floats (OGS, http://www.euro-argo.eu/Activities/Floats-Developments-Deployments/Existing-Floats).

**Moored instruments**

Finally, CMRE provided in total 6 moorings, named W1, M1, M2, M3, M4, M5:

- W1 is a *Datawell* Waverider buoy (http://www.datawell.nl)
- M1 consisted of the central mooring $M1_{CTR}$ and a sideways extending appendage $M1_{APP}$ floating at the sea surface (Fig. 3). $M1_{CTR}$ was equipped with an upward-looking 300-kHz ADCP mounted at a nominal depth of 100 m below the sea surface, a CTD probe at 1 m depth, and a meteorological buoy at the surface (Table 4). $M1_{APP}$ was connected by a 50 m long rope to $M1_{CTR}$ and extended to about 40 m depth in the vertical direction. 41 Starmon mini temperature recorders (http://www.star-oddi.com) were densely spaced along the vertical cable in order to record temperature with high vertical resolution. In addition, 4 RBR data loggers (https://rbr-global.com) were mounted on the cable for determining the actual depth of the Starmons.
- M2, M3, M4, and M5 were "traditional" oceanographic sub-surface moorings equipped with CTD probes and current meters. A diagram of the general design of such a mooring, in this case M3, is displayed in Fig. 4, and the vertical arrangement of measuring devices for all moorings (M1, M2, M3, M4, M5) is shown in Table 4.

## 3  Schedule

The experiments at sea were separated into 3 legs (Table 5). Leg 1 started on 6 June with the departure of *Alliance* and *Planet* from La Spezia (Italy) and Toulon (France), respectively, and was solely dedicated to oceanography. Both vessels conducted a so-called CTD *initialisation survey* which covered the entire observational domain at 10 km horizontal resolution. In addition, 11 gliders, 6 moorings, 17 surface drifters, and 2 sub-surface floats were deployed by *Alliance*. Leg 1 was finished at 00:00 UTC (*Universal Coordinated Time*) on 12 June and *Alliance* called port in Porto Torres (Sardinia) for the exchange of personnel.

During Leg 2 (12–20 June), all activities on *Alliance* were related to acoustic experiments which will not be discussed in this article. However during this period of time, some limited data sets from short-term glider missions (not included in Table 5), CTD casts, and underway measurements are available for oceanographic analyses. At the same time, *Planet* continued with casts by lowered CTD (lCTD) and uCTD and measurements by the CTD chain.

After a short port call in Oristano (Sardinia, *Alliance* only), for another exchange of personnel, both vessels had a rendezvouz for the inter-calibration of their CTD probes on 21 June. Afterwards, so-called *validation surveys* were conducted for about 40

h until 23 June, using the ScanFish (*Alliance*) and the CTD chain (*Planet*) simultaneously on parallel tracks. Thereafter, *Planet* finished her field work and headed towards Palma de Mallorca. *Alliance* remained another day in the area for the recovery of all gliders and moorings, concluding the experiment on 25 June in La Spezia.

## 4   Observations

The purpose of all observations was to satisfy the objectives defined in the Introduction, except for the optical stations and the deployments of the ARVOR-I floats which were conducted in the framework of another project and in support of the MedArgo programme (Poulain et al., 2007), respectively. In particular, CTD data from shipborne casts and gliders and the underway measurements were applied for the investigation of water masses and circulation patterns (Objective 1). For Objectives 2, 3, and 4, all CTD data were used, and the ScanFish data sets were employed for the validation of model forecasts. The oceano-

graphic and meteorological data of the M1 mooring served as the backbone for Objective 5, and new payloads for gliders (Objective 6) were tested on various gliders by means of passive acoustic monitoring systems and pH/pCO$_2$ sensors on glider *Fin* (see Table 3). The data from the CTD chain are still under investigation. The surface drifter trajectories were included in the Mediterranean drifter data base at http://nettuno.ogs.trieste.it/drifter/database_med, and the time series recordings on moorings M2–M5 have not yet been exploited.

**CTD casts**

In total, 256 lCTD casts were taken by both vessels (Fig. 5). The scheduled vertical extent of all casts was 1000 dbar or bottom depth (whatever was shallower) but a few casts especially at the western boundary of the obserational domain reached greater depths in order to characterise the deep water masses. Casts taken during the deployment and recovery of moorings always

reached the bottom, whereas those taken during the deployment and recovery of gliders were limited to 1000 dbar as the gliders could not measure deeper than this. Additionally, 23 CTD profiles down to 350 m were recorded for the calibration of the CTD chain, 36 profiles down to a depth of 550 m were obtained by the uCTD, and 5 profiles to about 150 m were carried out with a CTD probe equipped with optical sensors (Tables 2, 5).

For the calibration of the CTD probes, water samples were taken with the rosette sampler on *Planet* on 57 stations during

Leg 1 and on the inter-calibration station on 21 June. On the same station, samples were taken by *Alliance* as well, while no samples were taken on any other station. This was not considered necessary because the CTD probes were equipped with a double line of sensors for temperature and salinity.

A comparison of the casts taken during Leg 1 revealed slight differences of 0.001 S m$^{-1}$ between the conductivities measured on both vessels, where the conductivity recorded by *Alliance* is the lower value. This leads to differences of salinity

and potential density of 0.010 and 0.008 kg m$^{-3}$, respectively. Although this shift is fairly small it is above the expected level of accuracy. Hereupon, Knoll et al. (2015b) investigated this issue in greater detail but it could not be clarified which sensor caused the observed shift. Therefore, the authors recommended to add 0.001 S m$^{-1}$ to all CTD profiles recorded by *Alliance*, or to subtract the same value from the conductivities measured by *Planet* . The result of this procedure is illustrated in Fig. 6.

Here are shown maps of potential density at 990 m depth before and after the applied correction.

**Glider missions**

11 gliders were deployed on 8 and 9 June in 2 batches of 6 and 5 respectively, at the positions marked by the green circles in Fig. 7a, and afterwards were directed to their nominal tracks G01, …, G10. The distribution of the gliders on their assigned tracks was arranged in a way that at least every other track was occupied by a "deep" glider (pressure rating >650 dbar). Track G08 was occupied by 2 gliders (*Dora* and *Minke*) because *Dora* was a brand new glider equipped with a propeller drive and its performance should be compared with that of a conventional glider. All gliders except for *Clyde* and *Fin*, accomplished their tasks satisfactorily until recovery on 23 June (Fig. 7b) while commuting up to 5 times between the eastern and western end of their tracks. *Clyde* had to be recovered earlier than planned on 12 June because of a hardware malfunction and was never redeployed, and *Fin* had to be recovered on 10 June because of a software problem. However, *Fin* was redeployed successfully the following day. After recovery, no apparent biofouling was observed on any glider.

**CTD chain and ScanFish**

The CTD chain towed by *Planet* came into operation twice, 12–14 June (Fig. 8a), and 21–23 June in sync with the ScanFish towed by *Alliance* as shown in Fig. 8b. The meridional distance between the zonal tracks in Fig. 8b is 10 km, and the zonal tracks are aligned with the zonal CTD sections (Fig. 5).

**Optical stations**

Optical measurements were carried out daily around local noon on 7–11 June. Each station consisted of an ensemble of vertical profiles with the free-falling HyperPro and 1 cast down to about 150 m depth using the CTD probe equipped with optical sensors. For the position of the casts, see Fig. 9. Additional optical data are available from some gliders equipped with optical sensors and from the ScanFish.

**Drifters and floats**

During transit from La Spezia to the survey area, 1 ARVOR-I float was launched on 6 June near the eastern mouth of the Bonifacio Strait in the Tyrrhenian Sea in support of the Argo programme and without any direct relevance to REP14-MED. A further 17 surface drifters and 1 more ARVOR-I float were deployed 8–11 June in the observational domain. The deployment positions are shown in Fig. 9.

**Moorings**

Six moorings were deployed 8–11 June at the positions shown in Fig. 9. W1 and M1 were recovered on 20 June, and M2, M3, M4, M5 on 23 June.

### Underway measurements

On *Alliance*, temperature and salinity at 2.5 m depth were recorded continuously by a thermosalinograph during the entire survey. At the same time, the 75-kHz ADCP acquired the three-dimensional velocity field under the vessel down to a depth of 653 m with a vertical resolution of 16 m. Usable data of the vessel-mounted 150-kHz ADCP on *Planet* were obtained 10–23 June only at depths below 80 m. The profiling depth varied between about 180 m during daylight and 280 m during the night (Knoll et al., 2015a). For the tracks of the vessels see Fig. 10.

Continuous shipborne observations of meteorological parameters were recorded on both vessels. On *Alliance*, records of wind speed, wind direction, air temperature, air pressure, relative humidity, and wind gusts were made at 3 locations on the foredeck (starboard and port) and at the stern, whilst on *Planet*, wind speed and direction at different positions on the vessel, relative humidity, air temperature, air pressure, visibility, altitude of clouds, and rain were recorded with sensors at 9 different locations within a height of 10–26 m. The seawater temperature was measured at 6 m depth. Additional records of various meteorological parameters are available from the meteorological buoy on top of mooring M1.

## 5   Complementary data

### Weather and wave prediction models

For the entire experiment, weather analyses and forecasts of the COSMO (*Consortium for Small-Scale Modeling*) atmospheric model were made available by the Italian Weather Service *Centro Nazionale di Meteorologia e Climatologia Aeronautica* (CN-MCA) in 2 different setups, COSMO-ME and COSMO-IT. COSMO-ME covers the entire Mediterranean Sea with a horizontal resolution of 7 km and provided 72-h forecasts. COSMO-IT encompasses Italy and the adjacent waters at a high resolution of 2.2 km but the forecast range was only 24 h. In addition, CNMCA also provided forecasts of the NETTUNO wave model for the entire Mediterranean.

### Ocean circulation models

In order to assess the environmental conditions in the survey area and to support the efforts of the REP14-MED modeling community, the outputs of 5 ocean circulation models were downloaded routinely from the respective servers. Products of the *Mediterranean Forecasting System* (MFS) and the MERCATOR global model were provided via CMEMS (*COPERNICUS Marine Environment Monitoring Service*), WMED – a high-resolution model of the waters around Sardinia – was obtained from IAMC, WMOP (*Western Mediterranean Operational Forecasting System*) was produced by SOCIB, and finally the Extended Range Prediction System was provided by NRL.

### Remote sensing

Derived sea surface temperature was collected during the experiment by the CMRE *TeraScan* satellite system from the polar

orbiting satellites POES and METOP.

## Historical data

Historical data include temperature and salinity profiles extracted from the *World Ocean Database 2013* (Boyer et al., 2013) in a wider area surrounding the REP14-MED observational domain, and 4 CTD cruises conducted by IAMC 2001–2003 (Ribotti et al., 2004).

## 6 Discussion

This article describes the oceanographic surveys of the REP14-MED experiment. The description is restricted to the geographic and oceanographic settings of the survey area, the schedule, the employed measuring systems, the timing and the positioning of all observations, and the availability of complementary data. No observations are shown; this is to avoid the anticipation of potential results of forthcoming publications.

The surveys undertaken at sea provide an extremely rich dataset that can be used for multiple studies on different topics. The major advantage of the data set is its consistency, which was not affected by the handful of glitches experienced:

- Glider *Clyde* failed completely after 10 June. However, *Clyde* was fortunately positioned on track G01 and the malfunction did not impact the regularity of the entire glider track pattern.
- Glider *Fin* dropped out for about 1 day.
- On 23 June, *Planet* had to circumvent a fishing area while towing the CTD chain. This explains the northward excursion from track P09b in Fig. 8.
- Originally on mooring M1, a CTD probe was mounted directly at the sea surface but the recording of conductivity failed shortly after the deployment of the mooring.
- Two out of 40 thermistors in mooring M1 failed.

The objectives of the experiment as defined in the Introduction have been addressed by 9 scientific articles (including this one) which are assembled in a special issue of *Ocean Science*: using the data collected by the shipborne CTD probes, gliders, towed instruments, and vessel-mounted ADCPs, Knoll et al. (2017) investigated the hydrography and circulation patterns in the experimental area and compared them with previous knowledge. It turned out that the distribution of the water masses and the circulation resembled the classical picture as described by Millot (1999), but there were also significant differences: the temperature and salinity of MAW, LIW, and Bottom Water had increased compared to the observations during the last decade. By contrast to previous observations, LIW occupied the whole trial area; no LIW vein tied closely to the Sardinian coast was found south of 40° N. Within the MAW, an unusually cold and saline eddy was observed in the southern trial area while extremely low saline surface water entered the area in the southwest towards the end of the experiment. An anticyclonic WIW eddy was identified which may confirm the existence of a direct route of WIW from its formation region in the Ligurian Sea and in the Gulf of Lion to the observational site. Geostrophic transports were in reasonable agreement with the ADCP

measurements. Within the MAW, northward transport was observed on the shelf while the currents over the slope headed to the south. Northward transport of LIW was predominantly observed offshore.

Concerning operational forecasting, a relocatable ocean prediction system was applied in hindcast mode to the survey data (Onken, 2017a). More than 6000 temperature and salinity profiles originating from the glider fleet and shipborne CTD probes were assimilated, and the forecasts were verified against observations from the ScanFish tows (Fig. 8b). In a sensitivity study, the forecast skill score (as defined by Murphy (1993)) was determined in dependency of various assimilation parameters, and it was shown that the score increased with increasing forecast range. Moreover, it was demonstrated that the vast number of observations can be managed by the applied objective analysis method without data reduction, enabling timely forecasts with acceptable goodness even on a commercially available personal computer.

Different schemes were adopted to assimilate the survey data. While Onken (2017a, b) applied objective analysis, a hybrid variational-ensemble scheme was developed by Oddo et al. (2016), and Hernandez-Lasheras et al. (2018) used a local multi-model ensemble optimum interpolation (EnOI). The hybrid scheme of Oddo et al. (2016) corrects both the systematic errors introduced from the initialisation, lateral and surface boundary conditions, and model parameterisation, and improves the representation of small-scale errors in the background error covariance matrix. An ensemble system was run offline for further use in the hybrid scheme, generated through perturbation of the assimilated observations. The results of 4 different experiments were compared. While the reference experiment had no systematic error correction, the other 3 experiments accounted for error corrections, combining the static and the ensemble-derived errors of the day. It was shown that the hybrid scheme reduced the mean absolute errors of the temperature and salinity misfits by 55 and 42%, respectively, versus statistics arising from standard climatological covariances without systematic error correction. Hernandez-Lasheras et al. (2018) presented the results of different simulations, assimilating temperature and salinity profiles from shipborne CTDs and gliders from REP14-MED, along with observations of sea surface temperature, sea level anomaly and Argo profiles over the whole Western Mediterranean Sea. The objectives were to explore the performance of the EnOI data assimilation scheme in ingesting both large scale data and dense profile observations over a reduced area, and to compare the performance of the CTD initialisation survey versus different configurations of glider fleet sampling including up to 8 vehicles. Results show the capacity of the system to ingest both type of data, leading to improvements in the representation of all assimilated variables. These improvements persist during the 3-day periods separating two analyses. However, the system presents some limitations in properly representing the anticyclonic eddy observed around 50-m depth in the southern part of the domain, due to the reduced vertical extension of the eddy, which is smoothed out by the vertical model error covariances provided by the ensemble. A verification of the forecasts using independent measurements from shipborne CTDs and the Scanfish shows that the simulations assimilating initial CTD data reduce the error by 30 to 40% with respect to the simulation without data assimilation. In the glider-data-assimilative experiments, the forecast error is reduced in terms of the root-mean-square error as the number of vehicles increases. The simulation assimilating CTDs outperforms the simulations assimilating data from 1 to 4 gliders. A fleet of 8 gliders provides a similar performance as the 10-km spaced CTD initilisation survey in these experiments. Moreover (A. Funk, personal communication, 2018; not included in the special issue), the incremental strong constraint 4Dvar technique was applied. Both profiles from shipborne CTDs and gliders are assimilated, and the impact of different sets of observational data on prediction quality is estimated.

Convergence issues of the assimilation scheme are analysed and overall performance is compared to other data assimilation techniques.

A cost/benefit analysis of gliders vs. traditional platforms was presented by Russo et al. (2018): traditional observing platforms, i.e. research vessels, are expensive in terms of both human and financial resources, and are not able to provide comprehensive observations in challenging areas like the polar oceans. By contrast, observations on global scales are obtained from satellites but only for the sea surface. Hence, the underwater environment still remains largely unobserved. Autonomous underwater platforms promise to provide an affordable solution for filling the observative gap. In particular, marine gliders are demonstrating their capabilities for long missions by crossing the ocean basins up to increasing depths. Long endurances at sea are achieved at the cost of sacrificing glider speed, and this fact generates issues for several possible utilisations of the observations collected by gliders. A partial solution was represented by a fleet of coordinated gliders during the REP14-MED sea trial where 2 research vessels and 11 gliders sampled simultaneously the same limited area. This offered the opportunity for conducting a comparison between the 2 data sets aiming to assess several aspects, including capability to detect oceanographic features, improvements of operational forecasting skill, and economic costs.

The time series records of mooring M1 enabled an experiment for the exploration of the variability of the ocean. Onken (2017a) employed the Regional Ocean Modeling System (ROMS, Shchepetkin and McWilliams (2005)) in offline-nested mode to assess the sensitivity of the forecast skill of mixed-layer properties to initial conditions, boundary conditions, and mixing parameterisations. All forecasts were validated against the M1 thermistor records. The initial and lateral boundary conditions were provided by the Mediterranean Forecasting System (MFS, Tonani et al. (2014)) and by the MERCATOR global ocean circulation model (Drévillon et al., 2008). Nesting ROMS in MERCATOR and updating the initial conditions through data assimilation provided the best agreement of the predicted mixed-layer properties. Further improvement was obtained if the predicted atmospheric forcing fields were melded with real observations, and by the application of the k-$\omega$ vertical mixing scheme (Wilcox, 1988) with increased vertical eddy diffusivity. The predicted temporal variability of the mixed-layer temperature was reasonably well correlated with the observed variability, while the modelled variability of the mixed-layer depth exhibited only agreement with the observations near the diurnal frequency peak. For the forecasted horizontal variability, reasonable agreement was found with observations from a ScanFish section, but only for the mesoscale wave number band; the observed submesoscale variability was not reproduced.

New payloads for gliders were tested during the experiment: glider *Clyde* was equipped with 2 hydrophones and an experimental pH/$p$(CO$_2$) sensor was for the first time installed on glider *Fin*. Although *Clyde* failed about 36 hours after its deployment, the acoustic records were intact and complete because they were anyway limited to approximately 23 hours. The equipment used and the analysis of the data obtained are discussed by Küsel et al. (2017). Sperm whale regular clicks as well as dolphin clicks and whistles were identified. Cross-correlating clicks recorded on both hydrophones allowed for the estimation of the bearing of clicks and realisation of animal tracks. Two dual pH/$p$(CO$_2$) sensors were mounted on *Fin* – 1 standalone pair and another one integrated into the glider electronics. However, due to quality issues, only the data of the standalone pH sensor could be used and the article of Hemming et al. (2017) is restricted to that one. During the deployment, pH was sampled over a period of 12 days. The vertical resolution of the pH sensor was good, but stability was poor and the sensor drifted in a

non-monotonous fashion. An offset was applied to remove the sensor drift, reducing the spread of the data by approximately 2/3. Additional corrections decreased the apparent pH variability by a further 13 to 31%. Sunlight caused a sensor pH decrease in surface waters around local noon, highlighting the importance of shielding the sensor from light in future deployments. The corrected pH is presented along with other parameters measured by the glider. Maxima of pH were identified close to the depth of the summer chlorophyll maximum, and higher pH was associated with saltier LIW.

## 7 Conclusions

In June 2014 and in the framework of the REP14-MED experiment, a huge oceanographic dataset was collected by 2 research vessels, 11 gliders, and other measuring systems. The large volume of the dataset, its consistency and the quality, were the stimulus for various scientific studies which are published in the special issue "REP14-MED: A Glider Fleet Experiment in a Limited Marine Area" of this journal. This article provides an overview of the logistics of the survey and the observations at sea, and highlights the major findings of related articles published so far. All articles comply with the objectives defined above and address the following subjects:

– Water masses and circulation off the west coast of Sardinia;

– Operational forecasting and model validation;

– Different methods for data assimilation;

– Variability of the ocean;

– Cost/benefit analysis of gliders;

– New payloads for gliders.

In the publications of the special issue, most of the defined objectives were addressed, but the richness of the obervations has by far not yet been exploited adequately:

– The "wavelength" of the sawtooth-like glider path is between $\mathcal{O}(100\,\mathrm{m})$ and $\mathcal{O}(1\,\mathrm{km})$ depending on the water depth. Hence, the along-path resolution is on the same order of magnitude as that of submesoscale features and would enable corresponding analyses. Moreover, the alternating occupation of the tracks by deep and shallow gliders may enable model studies which assess the dynamical impact of the sub-thermocline flow on the overall circulation pattern. For instance, the data of all available gliders may be assimilated in a control run, while in another run the glider data below approximately 200 m depth are ignored. The huge amount of glider data also advances assimilation experiments aimed to test the efficiency of different sampling patterns (Oke et al., 2015) in terms of sub-sampling or "super observations" (Lorenc, 1981).

- The original intention of the the ScanFish surveys (Fig. 8) was to provide a quasi-synoptic validation data set for model forecasts. As the wavelength of the undulating device is around 1 kilometre (in deep water), the data are also suitable for investigations of submesoscale processes.

- The temporal resolution of the CTD chain is extremely high at about 3 s which is equivalent to a spatial resolution of less than 10 m at a tow speed of 4–5 kn. This would allow very detailed studies of submesoscale features in the transition regime to the microscale.

- The sub-surface moorings M2, M3, M4, and M5 were equipped with CT probes and current meters (partly upward looking ADCPs) at standard depths. Hence, time series of scalar parameters and velocities are available for the upper ocean, the diurnal and seasonal thermocline, and for the levels of the intermediate and deep water masses. They may be
used as well for model validation and provide a deeper insight into the meridional exchange of water masses across the 40° N latitude circle.

Hopefully, this article may encourage forthcoming studies which address these issues.

*Data availability.* All data of the REP14-MED experiment are available on the CMRE ftp server at ftp.cmre.nato.int. Requests
for access may be directed to registry@cmre.nato.int or pao@cmre.nato.int.

*Author contributions.* Reiner Onken was the coordinator of the experiment and Chief Scientist on NRV *Alliance*, Heinz-Volker Fiekas was Chief Scientist on RV *Planet*, Laurent Beguery was the Principal Investigator from ALSEAMAR and provided the SeaExplorers, Ines Borrione and Aniello Russo were responsible for the collection and processing of CTD and ADCP data on NRV *Alliance*, Karen Heywood and Jan Kaiser were the Principal Investigators from UEA and provided the SeaGliders, Pierre-Marie Poulain provided the SVP drifters and
20 the ARVOR-I floats, Michaela Knoll and Andreas Funk collected and processed all data on RV *Planet* and the data from the WTD71 glider, Bastien Queste and Michael Hemming processed the SeaGlider data, Kiminori Shitashima was in charge of the new pH/pCO$_2$ sensor on glider *Fin*, Jaime Hernandez-Lasheras, Baptiste Mourre, and Paolo Oddo contributed their assimilation studies, and Martin Siderius was the Principal Investigator from PSU; he provided the Slocum glider *Clyde* which was piloted by Elizabeth Thorp Küsel.

*Acknowledgements.* The authors would like to thank all partners which made REP14-MED such a great success. We appreciate the hard work
of the officers, scientists, engineers and technicians on NRV *Alliance*, RV *Planet* and ashore who were responsible for the logistics of the experiment, launch and recovery of instruments, piloting of gliders, data acquisition, processing and archiving: Marion Mery (ALSEAMAR), Marina Ampolo-Rella, Peter Anger, Gisella Baldasserini, George Botezatu, Alessandro Brogini, Daniele Cecchi, Giampolo Cimino, Marco Colombo, Rodney Dymond, Silvia Falchetti, Bartolome Garau, Craig Lewis, Michele Micheli, John Osler, Ivan Pennisi, Giuliana Pennucci, Richard Stoner, and Alex Trangeled (CMRE), Giovanni Quattrocchi and Andrea Satta (IAMC), Stefan Riedel (MARKDO), Gareth Lee and
Stephen Woodward (UEA), Jens Benecke, Evelyn Ehlers, Malte Ehlers, Jörg Förster, Stefan Hasenpusch, Bernd Hilgenfeld, Daniel Jähne, Ralf Lühder, Lars Planert, Horst Schock, Marco Weidemann, and Lukas Zimmermann (WTD71). We thank *Kongsberg Maritime* who loaned SeaGlider *Kong* to UEA for this experiment. A special thanks is adressed to the masters and crews of NRV *Alliance* and RV *Planet* for their

professionalism during the conduction of the experiments at sea, and to AMP who agreed to support the recovery of gliders in emergency situations. REP14-MED was sponsored by HQ Supreme Allied Command Transformation (Norfolk, VA, USA), and Michael Hemming's PhD project analysing REP14-MED data was funded by the Defence Science and Technology Laboratory (DSTL, UK).

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

**Table 1.** Partners of the experiment

| Acronym | Name | Country |
| --- | --- | --- |
| ALSEAMAR | Architecture et Conception de Systemas Avantes | France |
| AMP | Area Marina Protetta Peninsola del Sinis | Italy |
| CMRE | Centre for Maritime Research and Experimentation | NATO |
| DGA | Direction Générale de l'Armement | France |
| DSTL | Defence Science Technology Laboratory | United Kingdom |
| HLS | Heat, Light and Sound Research | United States of America |
| IAMC | Istituto per l'Ambiente Marino Costiero | Italy |
| IIM | Istituto Idrografico della Marina | Italy |
| ISMAR | Istituto di Scienze Marine | Italy |
| MARKDO | Marinekommando | Germany |
| MIT | Massachusetts Institute of Technology | United States of America |
| MWC | Maritime Warfare Centre | United Kingdom |
| NRL | Naval Research Laboratory | United States of America |
| OGS | Istituto Nazionale di Oceanografia e di Geofisica Sperimentale | Italy |
| ONR | Office of Naval Research | United States of America |
| SOCIB | Sistema de Observación Costero de las Islas Balears | Spain |
| PSU | Portland State University | United States of America |
| TWR | Teledyne Webb Research | United States of America |
| UEA | University of East Anglia | United Kingdom |
| UPMC | Université Pierre et Marie Curie | France |
| WTD71 | Wehrtechnische Dienststelle für Schiffe und Marinewaffen, Maritime Technologie und Forschung | Germany |

**Table 2.** Summary of all observations. The variables names $p, pCO2, pH, T, S, U, V$ refer to pressure, $CO_2$ partial pressure, pH value, temperature, salinity, zonal and meridional velocity, respectively. Only those variables are mentioned which are relevant for the articles in the special issue.

$^*$ Additional parameters are available from these systems but they are beyond the scope of this article.

| Instrument | Data type | Variables | June Period | Track length [km] | Data volume | Remarks |
|---|---|---|---|---|---|---|
| Gliders | 3D trajectory | $T, S, p, pH, pCO2^*$ | 08–23 | 3278 | 133 glider days 5731 profiles | |
| ScanFish | 3D trajectory | $T, S, p^*$ | 21–23 | 365 | 535 profiles | resolution < 1321m |
| Thermo-salinograph | 2D trajectory | $T, S$ @2.5 m depth | 06–25 | 3158 | | resolution < 10m |
| Surface drifters | 2D trajectory | $U, V, T$ | 08–25 | | 17 drifters | 6-h sampling interval |
| Floats | 2D trajectory | $U, V, T, S$ | 08–25 | | 1 float | within observational domain |
| | vertical profile | $T, S$ | 08–28 | | 3 profiles | within observational domain |
| lCTD casts | vertical profile | $T, S, p^*$ | 07–24 | | 279 profiles | spatially crowded on Leg 2 |
| uCTD casts | vertical profile | $T, S, p$ | 11-16 | | 36 profiles | |
| CTD chain | vertical profile | $T, S, p$ | 11–14 21–23 | 722 | > 100 000 profiles | resolution < 10m |
| ADCP (ship) | vertical profile | $U, V$ | 07–25 | 5561 | > 400 000 profiles | |
| Moorings | time series | $T, S, U, V^*$ | 08–23 | | | |
| Meteorol. obs. | time series | | 06–25 | | | shipborne and on M1 |

**Table 3.** Specifications of the gliders provided for the experiment. *Greta*, *Natalie*, and *SEA004* were deployed only during Leg 2 within the framework of the acoustic experiments, *Laura* and *Sophie* served as backup. For the specifications of most sensors, see the web sites of the manufacturers in Section 2. *PAM=passive acoustic monitoring system

| name | make | rating [dbar] | operator | Sensors and special properties |
|------|------|--------------:|----------|--------------------------------|
| Clyde | Slocum | 200 | PSU | CTD, PAM* |
| Dora | Slocum | 1000 | CMRE | CTD (pumped), propeller |
| Elettra | Slocum | 200 | CMRE | 2×CTD (pumped and unpumped) |
| Greta | Slocum | 200 | CMRE | CTD, PAM* |
| Jade | Slocum | 1000 | CMRE | CTD (pumped), propeller |
| Laura | Slocum | 200 | CMRE | CTD |
| Natalie | Slocum | 200 | CMRE | CTD, PAM* |
| Noa | Slocum | 1000 | CMRE | CTD, *Wetlabs* puck |
| Sophie | Slocum | 200 | CMRE | CTD, 504 nm irradiance, *Wetlabs* puck |
| WTD71 | Slocum | 200 | WTD71 | CTD (pumped) |
| Zoe | Slocum | 200 | CMRE | CTD, 504 nm irradiance, *Wetlabs* puck |
| Fin | Seaglider | 1000 | UEA | SBE CT sail, $O_2$, 2×pH/$pCO_2$ |
| Minke | Seaglider | 1000 | UEA | SBE CT sail, $O_2$ |
| Kong | Seaglider | 1000 | UEA | SBE CT sail, PAM* |
| SEA003 | SeaExplorer | 650 | ALSEAMAR | GPCTD, $O_2$, Chl A, CDOM, Bs770nm |
| SEA004 | SeaExplorer | 650 | ALSEAMAR | PAM* |

**Table 4.** Details of the moorings. All times are UTC. *RCM* stands for the Aanderaa current meters of the RCM series (http://www.aanderaa.com)

| | W1 | M1 | M2 | M3 | M4 | M5 |
|---|---|---|---|---|---|---|
| Longitude | 08°16.44′E | 08°16.22′E | 08°12.98′E | 07°57.65′E | 07°49.68′E | 07°35.39′E |
| Latitude | 39°30.71′N | 39°30.80′N | 40°05.98′N | 40°06.14′N | 40°06.36′N | 40°05.24′N |
| Water depth | 150 | 150 | 150 | 700 | 1300 | 2100 |
| Deployment | 08 June 07:47 | 08 June 07:14 | 10 June 06:42 | 10 June 11:30 | 10 June 15:56 | 11 June 07:18 |
| Recovery | 20 June 13:28 | 20 June 13:55 | 23 June 14:10 | 23 June 11:47 | 23 June 08:45 | 23 June 06:06 |
| 0 m | Waverider | Meteo buoy | | | | |
| 1 m | | CTD | | | | |
| 20 m | | | | CTD | CTD | CTD |
| 100 m | | | | CTD | CTD | CTD |
| | | ADCP 300 kHz | ADCP 300 kHz | | | ADCP 300 kHz |
| 200 m | | | | CTD | CTD | CTD |
| | | | | ADCP 150 kHz | ADCP 150 kHz | RCM |
| 500 m | | | | CTD | CTD | CTD |
| | | | | RCM | RCM | RCM |
| 1000 m | | | | | CTD | CTD |
| | | | | | RCM | RCM |
| 1500 m | | | | | | CTD |
| | | | | | | RCM |

**Table 5.** The observation schedules of *Alliance* and *Planet* and data availability of deployed systems.

| June date | 6 | 7 | 8 | 9 | 10 | 11 | 12 | 13 | 14 | 15 | 16 | 17 | 18 | 19 | 20 | 21 | 22 | 23 | 24 | 25 | Total |
|---|---|---|---|---|---|---|---|---|---|---|---|---|---|---|---|---|---|---|---|---|---|
| | | | Leg 1 | | | | | | Leg 2 | | | | | | | | | Leg 3 | | | Total |
| *NRV Alliance* | | | | | | | | | | | | | | | | | | | | | |
| lCTD casts | | 10 | 8 | 8 | 11 | 14 | | 1 | 11 | 5 | | 10 | 14 | 4 | 1 | 1 | | 5 | 3 | | 106 |
| Optical stations | | 1 | 1 | 1 | 1 | 1 | | | | | | | | | | | | | | | 5 |
| Deployment of gliders | | | 6 | 5 | | | | | | | | | | | | | | | | | 11 |
| Deployment of moorings | | | 2 | | 3 | 1 | | | | | | | | | | | | | | | 6 |
| Deployment of drifters | | 7 | 5 | 2 | 2 | 1 | | | | | | | | | | | | | | | 17 |
| Deployment of floats | | | 1 | | | | | | | | | | | | | | | | | | 1 |
| ScanFish tows | | | | | | | | | | | | | | | | xxxxxxxxxx | | | | | |
| Recovery of moorings | | | | | | | | | | | | | | | | 2 | | 4 | | | 6 |
| Recovery of gliders | | | | | 1 | | | | | | | | | | | | | 10 | | | 11 |
| Thermosalinograph | | xxxxxxxxxxxxxxxxxxxxxxxxxxxxxxxxxxxxxxxxxxxxxxxxxxxxxxxxxxxxxxxxxxxxxxxxxxxxxxxxxxxxxxxxxxxxx | | | | | | | | | | | | | | | | | | | |
| Shipborne ADCP | | xxxxxxxxxxxxxxxxxxxxxxxxxxxxxxxxxxxxxxxxxxxxxxxxxxxxxxxxxxxxxxxxxxxxxxxxxxxxxxxxxxxxxxxxxxxxx | | | | | | | | | | | | | | | | | | | |
| Ship meteorology | | xxxxxxxxxxxxxxxxxxxxxxxxxxxxxxxxxxxxxxxxxxxxxxxxxxxxxxxxxxxxxxxxxxxxxxxxxxxxxxxxxxxxxxxxxxxxx | | | | | | | | | | | | | | | | | | | |
| *RV Planet* | | | | | | | | | | | | | | | | | | | | | |
| lCTD casts | | | 8 | 17 | 19 | 16 | | | | | 10 | 18 | 19 | 18 | 17 | 8 | | | | | 150 |
| uCTD casts | | | | | | 2 | 6 | | 5 | 17 | 6 | | | | | | | | | | 36 |
| CTD chain tows | | | | | | | xxxxxxxxxx | | | | | | | | | xxxxxxxxxx | | | | | |
| lCTD (chain calibration) | | | | | | | 10 | | 1 | | | | | | | 7 | | 5 | | | 23 |
| Shipborne ADCP | | | xxxxxxxxxxxxxxxxxxxxxxxxxxxxxxxxxxxxxxxxxxxxxxxxxxxxxxxxxxxxxxxxxxxxxxxxxxxxxxxxxxxxxxx | | | | | | | | | | | | | | | | | | |
| Ship meteorology | | xxxxxxxxxxxxxxxxxxxxxxxxxxxxxxxxxxxxxxxxxxxxxxxxxxxxxxxxxxxxxxxxxxxxxxxxxxxxxxxxxxxxxxxxxxxxx | | | | | | | | | | | | | | | | | | | |
| *Data availability of deployed systems* | | | | | | | | | | | | | | | | | | | | | |
| Gliders | | | xxxxxxxxxxxxxxxxxxxxxxxxxxxxxxxxxxxxxxxxxxxxxxxxxxxxxxxxxxxxxxxxxxxxxxxxxxxxxxxxxxxxxxx | | | | | | | | | | | | | | | | | | |
| Moorings W1, M1 | | | xxxxxxxxxxxxxxxxxxxxxxxxxxxxxxxxxxxxxxxxxxxxxxxxxxxxxxxxxxxxxxxxxxxx | | | | | | | | | | | | | | | | | | |
| Moorings M2–M5 | | | | xxxxxxxxxxxxxxxxxxxxxxxxxxxxxxxxxxxxxxxxxxxxxxxxxxxxxxxxxxxxxxxxxxxx | | | | | | | | | | | | | | | | | |
| Surface drifters | | xxxxxxxxxxxxxxxxxxxxxxxxxxxxxxxxxxxxxxxxxxxxxxxxxxxxxxxxxxxxxxxxxxxxxxxxxxxxxxxxxxxxxxxxx | | | | | | | | | | | | | | | | | | | |
| Sub-surface float | | | xxxxxxxxxxxxxxxxxxxxxxxxxxxxxxxxxxxxxxxxxxxxxxxxxxxxxxxxxxxxxxxxxxxxxxxxxxxxxxxxxxxxxxx | | | | | | | | | | | | | | | | | | |

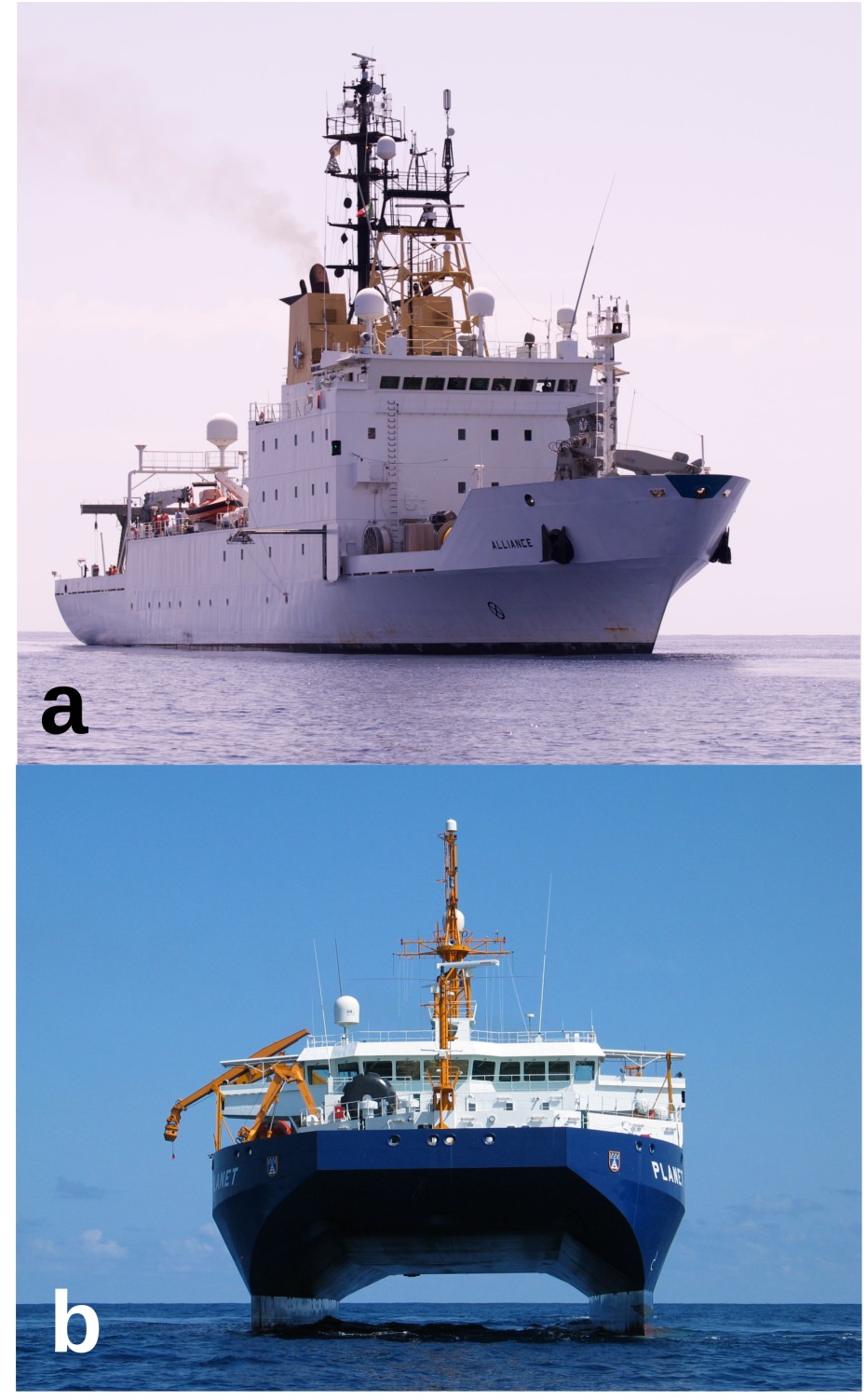

**Figure 1.** NRV *Alliance* (a) and RV *Planet (b)*

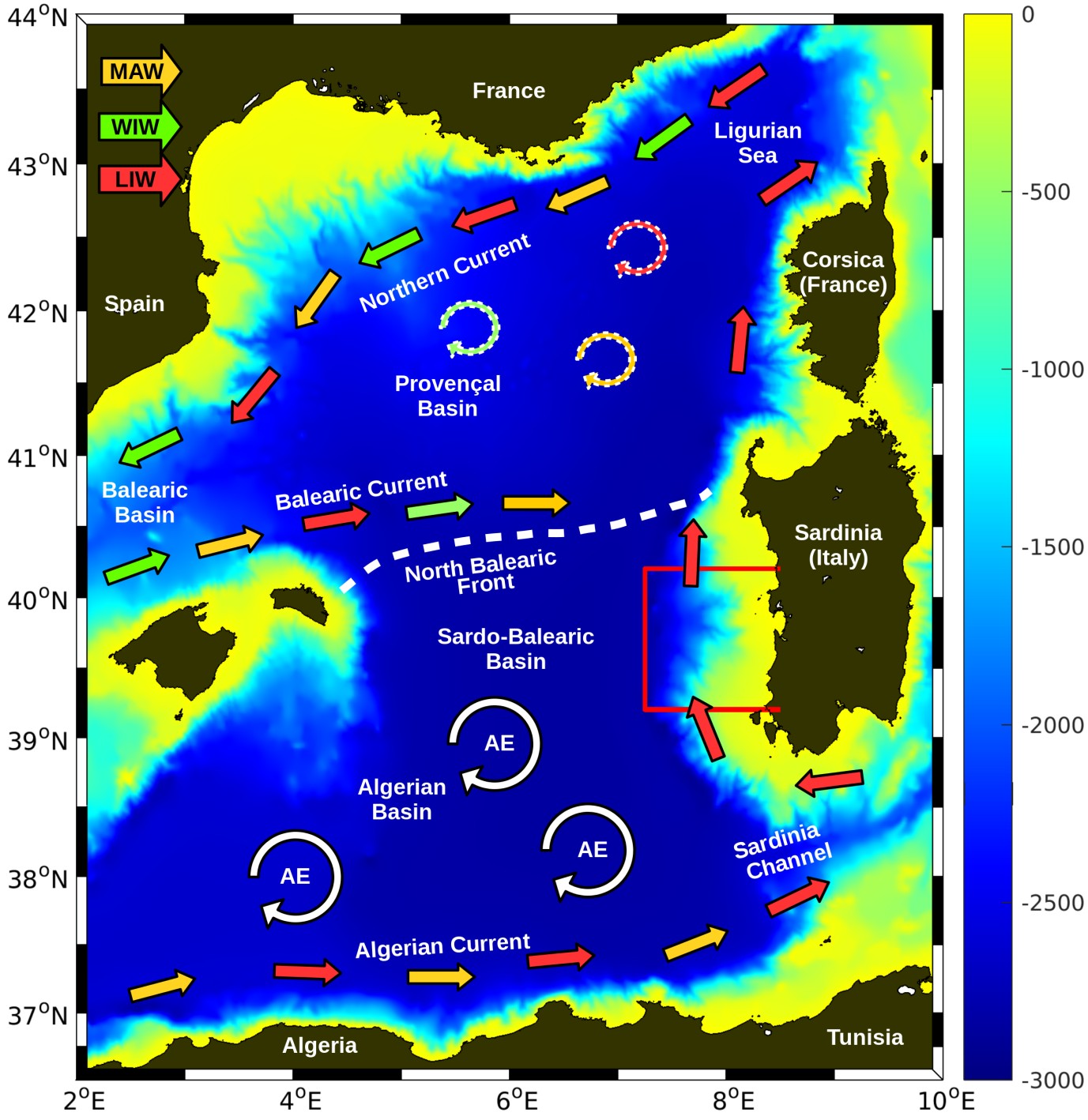

**Figure 2.** The large-scale circulation of the Western Mediterranean as it was known before the REP14-MED experiment. White-dashed circles indicate notional mesoscale eddies. The red box is the observational domain. Algerian eddies are marked by AE. For the other acronyms see text.

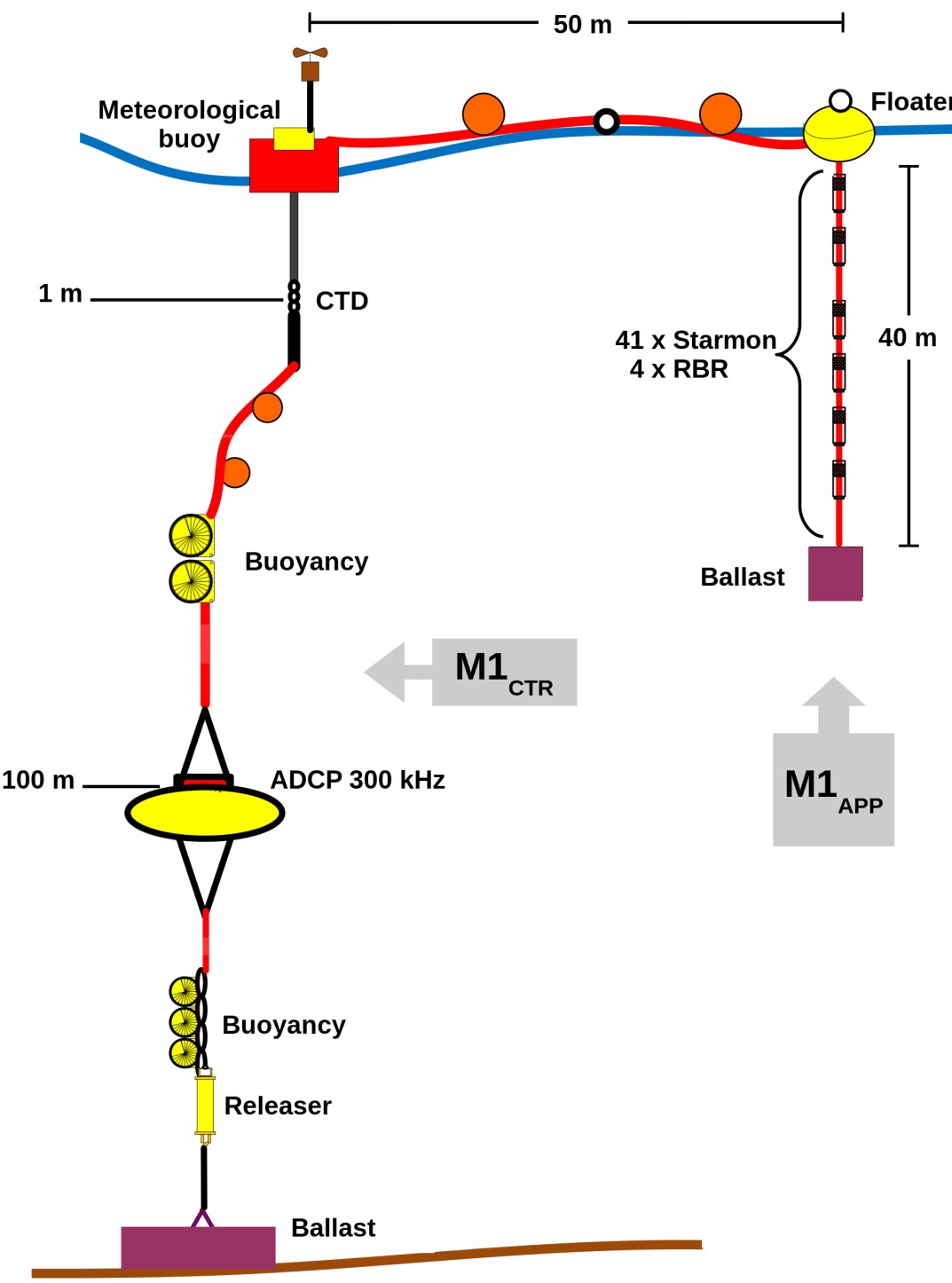

**Figure 3.** Design drawing of mooring M1. For explanations see text.

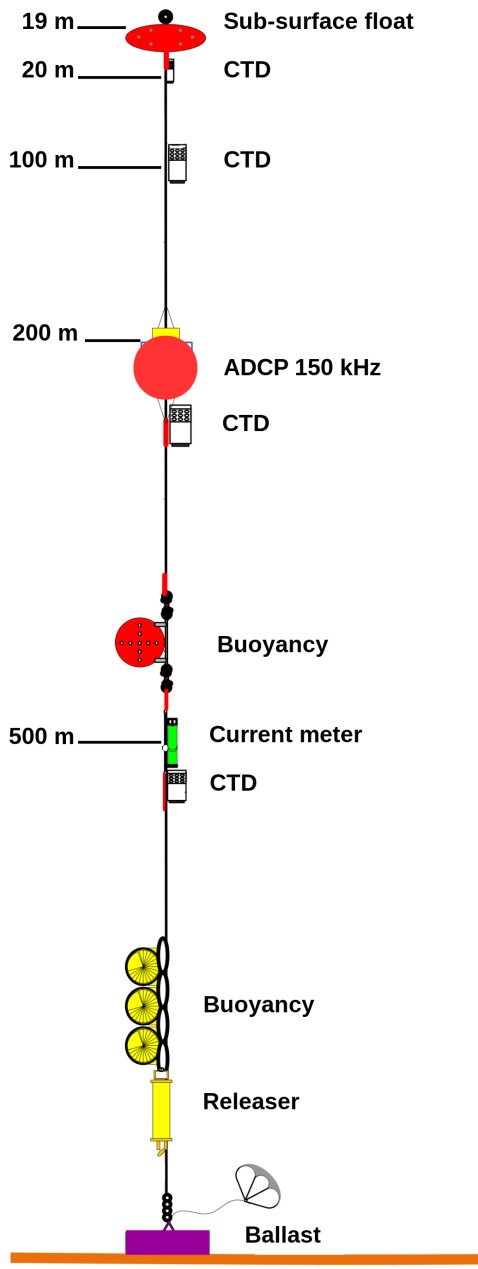

**Figure 4.** Design drawing of mooring M3. For explanations see text.

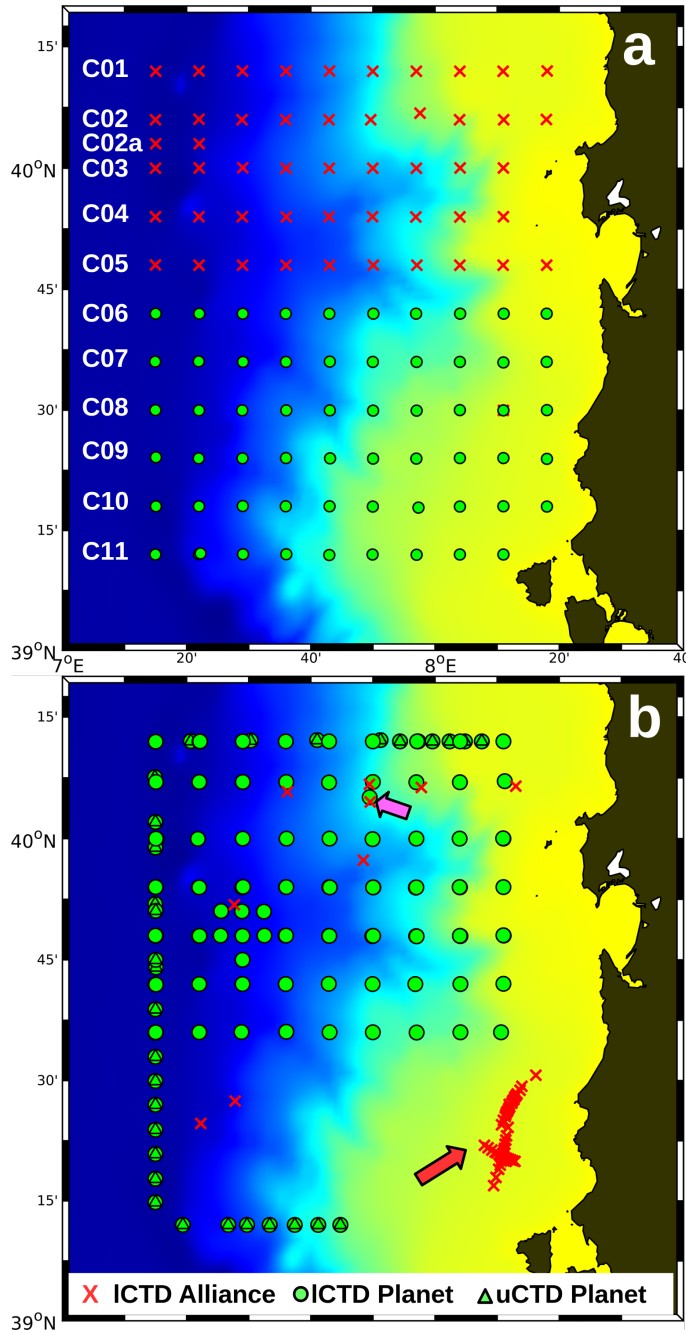

**Figure 5.** CTD casts taken during (a) Leg 1 and (b) Legs 2 and 3. Zonal sections in (a) are numbered C01, …, C11. The 46 CTD casts taken by *Alliance* in the framework of the acoustic experiments during Leg 2 are highlighted by the red arrow, the magenta arrow points to the position of the CTD inter-calibration. Extra casts taken by *Planet* for the calibration of the CTD chain are not shown. The colour code for the water depth is the same as in Fig. 2.

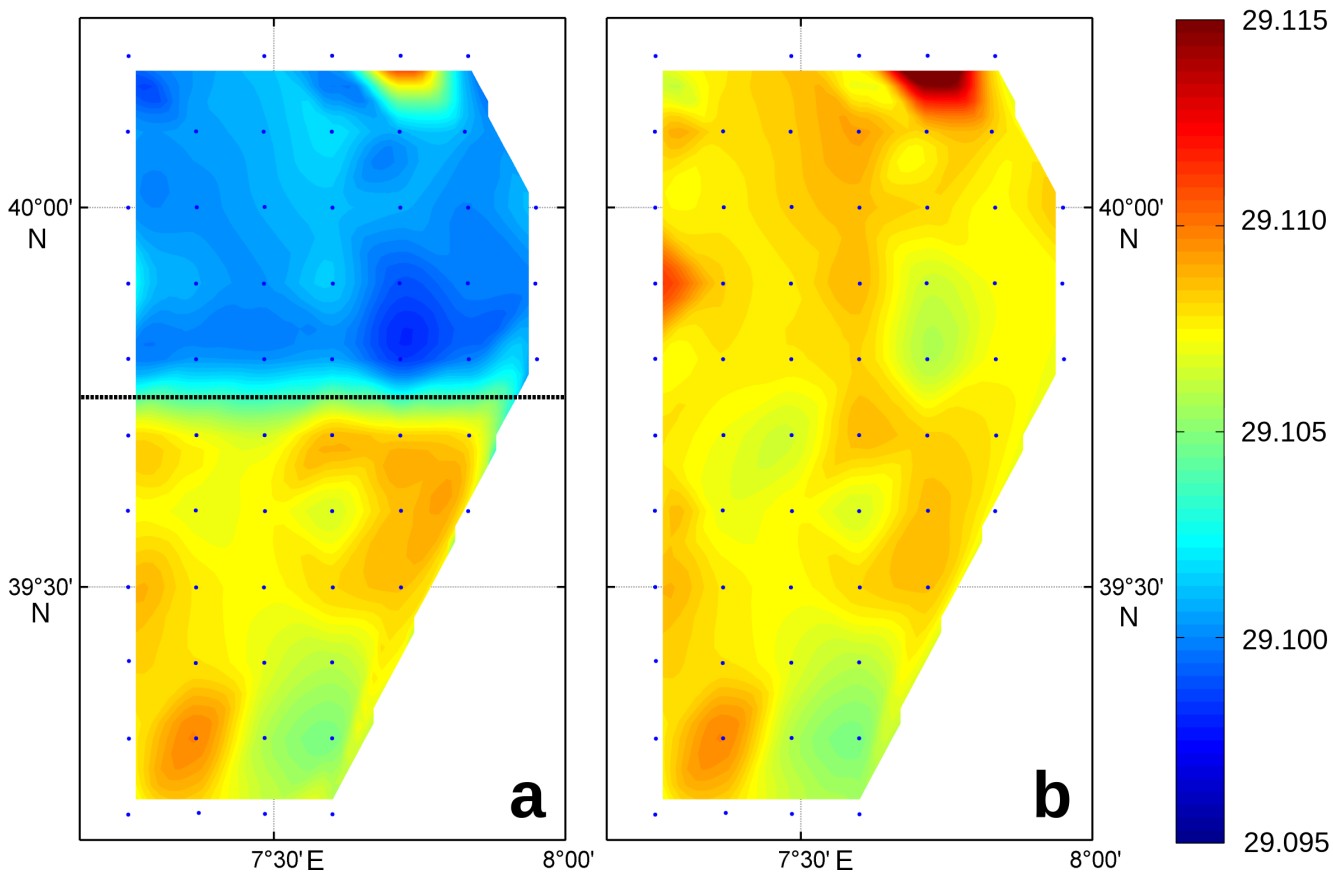

**Figure 6.** Potential density [kg m$^{-3}$] at 990 m depth (a) before and (b) after applying the salinity correction to the CTD casts taken during Leg 1. The positions of all casts are indicated by blue dots. All casts north of the dotted line were taken by *Alliance*, all casts south of that line were from *Planet*.

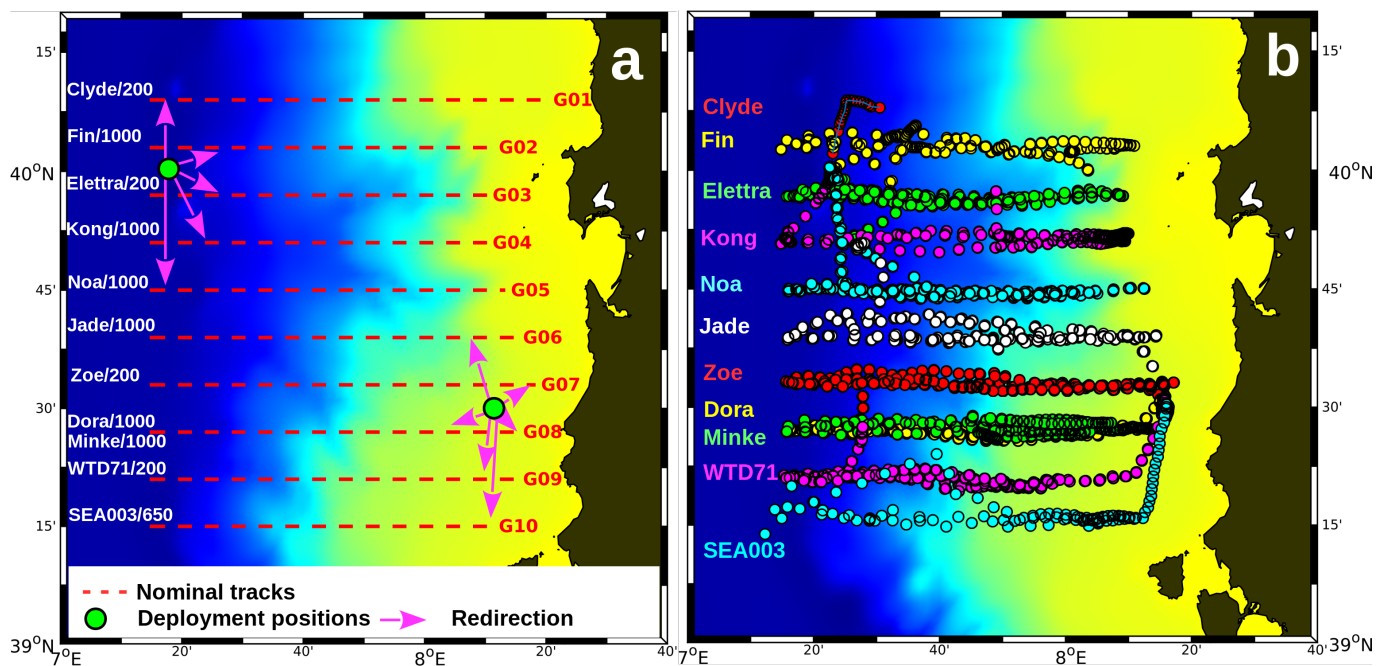

**Figure 7.** (a) Nominal tracks and deployment positions of all gliders launched during Leg 1. The names of the gliders are written at the western end of their respective tracks, and the number behind the name stands for the pressure rating in dbar. The gliders are redirected from the deployment positions to their nominal tracks. (b) Real tracks of all gliders 8–23 June. The circles indicate the surfacing positions. The colour code for the water depth is the same as in Fig. 2.

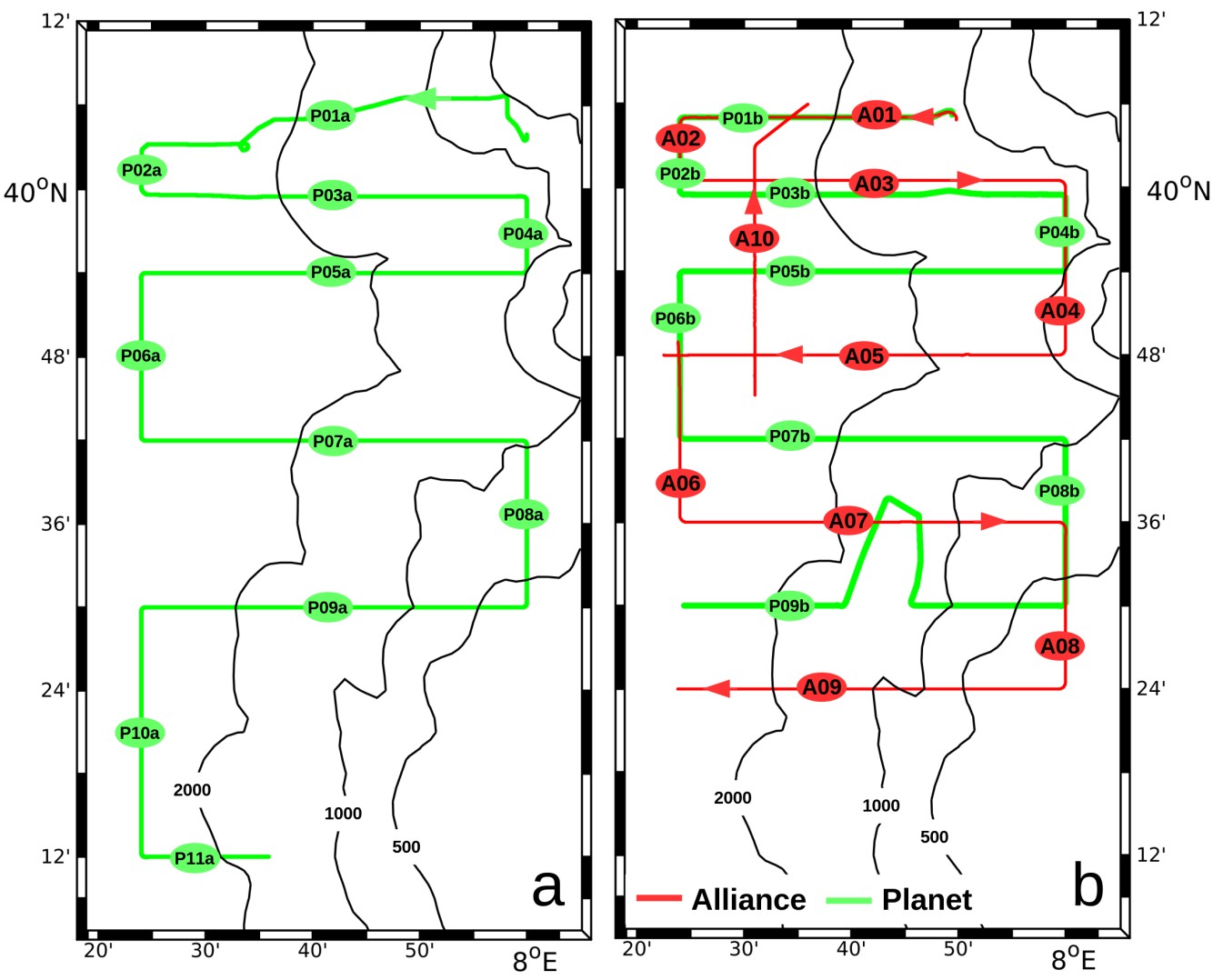

**Figure 8.** (a) CTD chain track of *Planet* during Leg 2. Tracks are numbered consecutively P01a, ..., P11a. (b) CTD chain tracks P01b,..., P09b of *Planet* and ScanFish tracks A01, ..., A10 of *Alliance* during Leg 3. Black lines indicate water depth [m].

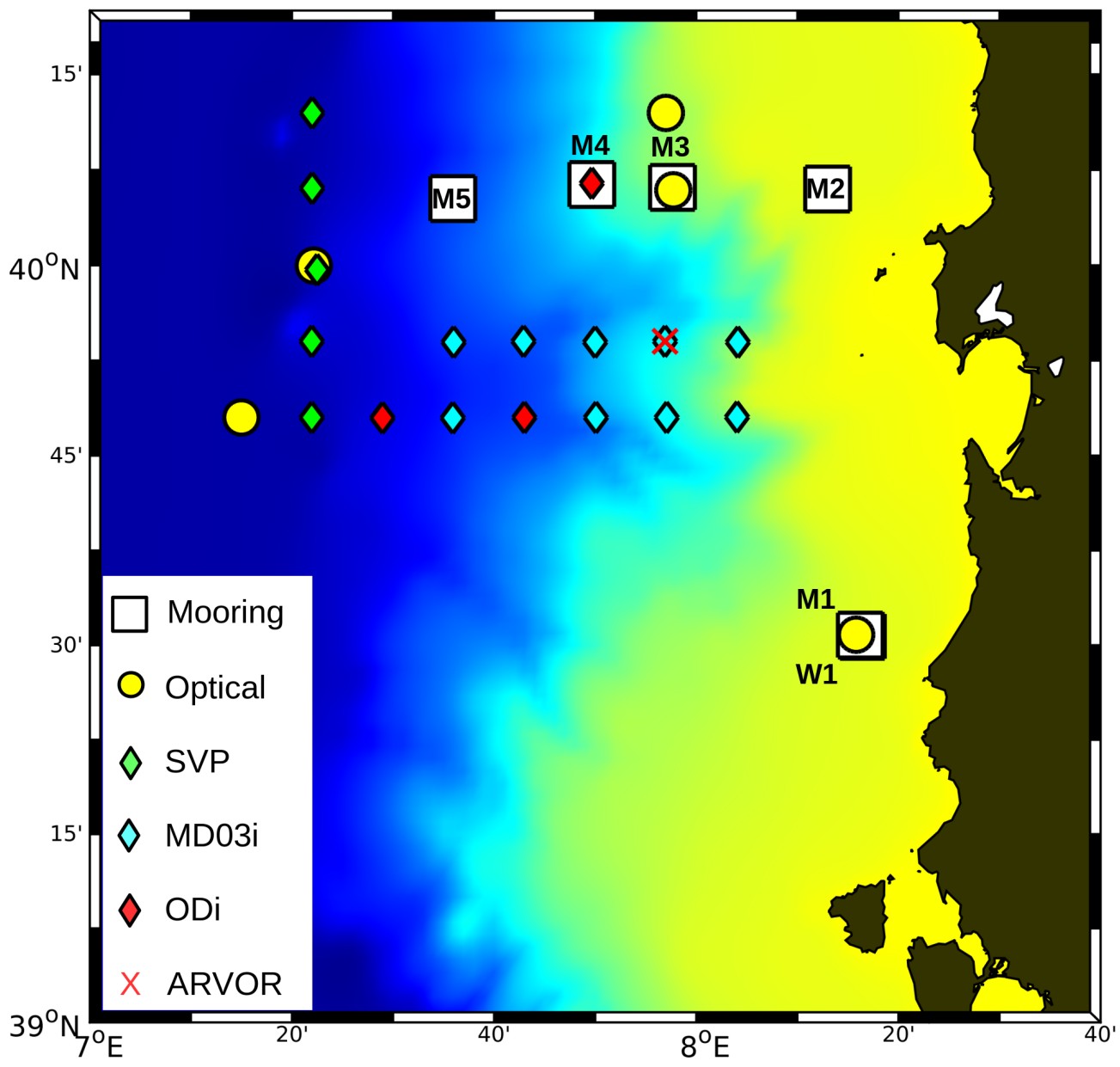

**Figure 9.** Positions of moorings, optical stations, and deployment positions of surface drifters (SVP, MD03i, ODi are drifter types) and the ARVOR-I float. The colour code for the water depth is the same as in Fig. 2.

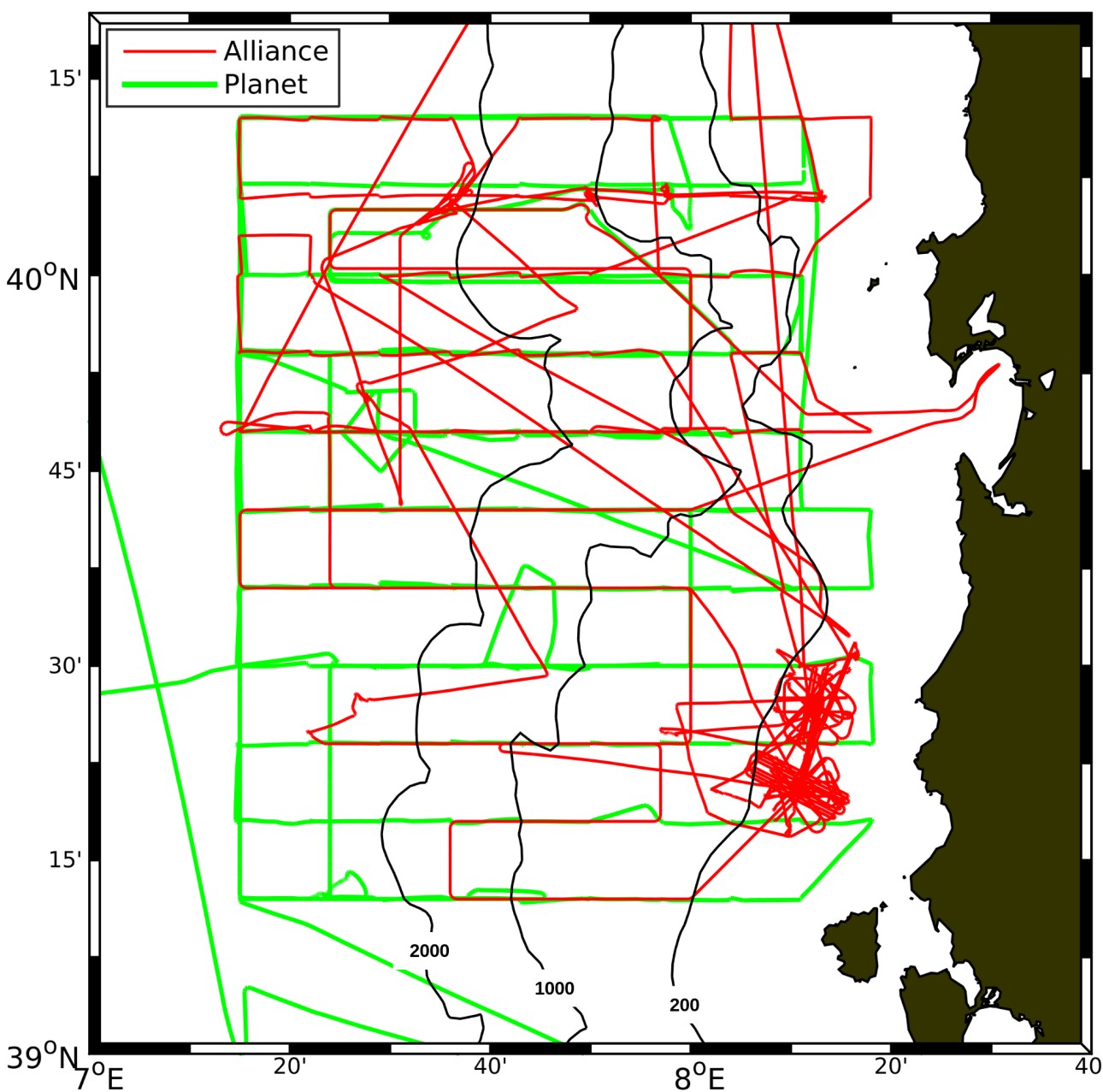

**Figure 10.** Ship tracks of *Alliance* and *Planet*. Black lines indicate water depth [m].