# Peer review of "High-Resolution Observations in the Western Mediterranean Sea: The REP14-MED Experiment"

_Ocean Science, 2016_

## Referee Comment (RC1) · Anonymous Referee #1 · 21 Nov 2016

The paper submitted here is, I guess, the introductory paper of the special issue "EP14-MED: A Glider Fleet Experiment in a Limited Marine Area". It describes the observational setting of the experiment, the reasons behind the selection of the region, and very few results about the calibration and intercalibration of the different sensors (quite exclusively limited to the ship based CTDs). The paper is well written and clear, although the aims are really limited. Not so much can be decrypted about the results of the experiment from the present text, as, I guess, others papers in the special issue will better detail the scientific advancements related to the REP-14 MED.

In my opinion, then, it is more a decision of the editor than of the reviewer, to publish or not that paper.

[Figure]

I have, anyway, two main suggestions/issues: 1. if the paper is the introductory paper for a special number, I would like to see at least some indications on the others papers of the special issue, with some, very rapid mentions of the mains results; I guess that authors have already some idea of papers that will be submitted and of their main results. 2. I cannot understand as, in the present "Argo era", temperature and salinity observations are still not publicly shared (as the authors indicated at the end of the draft) and they are "available only for the partners of the experiment". Although "interested institutions can sign up for partnership at any time", I considered that in the present day ocean science strategy would impose a more open data availability. But it is simply my opinion.

---

## Author Comment (AC1) · 22 Nov 2016

letter

**(1) The paper submitted here is, I guess, ..., REP14-MED ...**
That is true.

**(2) Very few results about the calibration ...**
The backbones of the REP14-MED survey were the ship based CTD casts and the glider data. Originally, it was planned to have an intercalibration station of the ship based CTD at the very beginning of the experiment in deep water, but because of a medical emergency situation on one of the research vessels the intercalibration had

to be postponed towards the end of the survey. Unfortunately, the water depth there was not sufficient for a meaningful comparison. An intercomparison of the gliders was never planned: eleven gliders were in use for the oceanographic experiment (plus 5 backup gliders), and they were from three different manufacturers (Teledyne, Kongsberg, ACSA) and owned by five different institutions (CMRE, UEA, ACSA, WTD71, PSU). A real intercomparison would have been too complex in terms of organisational issues; therefore, we relied on the calibration of the manufacturers.

**(3) The paper is well written ... the aims are really limited**
It was the intention of the authors, not to anticipate any results which might be subject of the other papers in the special issue.

**(4) ... it is more a decision of the editor ... to publish or not that paper**
I agree. Originally, the authors intended to include the description of the REP14-MED experiment in the Editorial of the special issue but that would have inflated the Editorial too much (11 pages text + 9 figures + 5 tables). Hence, it was decided to make the description of the experiment an independent paper which can be cited by other authors contributing to the special issue.

**(5) I would like to see ... indications on the other papers ... main results**
An overview of the other papers and their main results will be given in the Editorial of the special issue which will be written not before all papers have been accepted. Alternatively, I would suggest to hold up the present manuscript until the end of the submission window (actual deadline is 31 May 2017), and then include an overview of the other (accepted) papers and their main results. However, this would make the Editorial partly redundant or lead to overlapping of the Editorial and the present paper.

**(6) Temperature and salinity observations not publicly shared ...**
REP14-MED was an experiment under the lead of CMRE which is a NATO institution. Regrettably, the NATO data policy does not permit that any data are Open Access from the outset. However, this concerns only those data whereof NATO is the originator

(here, the data collected by CMRE). The other partners of the experiment are free to decide whether they want to make their data publicly available. Please note that Reiner Onken (the first author and Chief Scientist of the experiment) was affiliated with CMRE until 30 September 2016.

---

## Referee Comment (RC2) · Anonymous Referee #2 · 10 Jan 2017

**Review of the paper entitled:**

High-Resolution Observations in the Western Mediterranean Sea: The Rep14-MED Experiment

R. Onken et al.

Manuscript Number: OS-2016-82

The manuscript introduces the observational part of the REP14-MED experiment and gives a detailed description of measurements and instruments used for this experiment. REP14-MED is indeed an impressive project with a huge amount of different measurement strategies, instruments and scientists involved and it is therefore absolutely worthwhile to be published. Unfortunately, the authors remain shallow in their discussion and the manuscript is written like a technical report without any scientific aspects included at all. The authors emphasize that this article is designed for being an introduction to a Special Issue of Ocean Science and that they therefore do not want to present results of the observations. I do understand this but however I would expect more from this article or any article published in a scientific journal. There are several scientific aspects which can be discussed in such an introduction like:

- What is the overall scientific goal of the project?
- For what scientific reason were special measurements carried out? Scientific questions to be answered?
- What is the scientific goal of the special issue and which questions should be answered?
- What is the scientific content of each single manuscripts of the special issue (describing the results of the measurements)?

To my opinion none of these questions were discussed anyhow. Of course, the authors did a lot of honorable work and it is absolutely important to have such continuous measurements published, but however, a publication must be a scientific work and not only a presentation of data and instruments. For this reason I must reject the manuscript in its present form but I think by including a basic scientific discussion the article would be improved so that it can be presented once again.

---

## Author Response (AR1)

**Manuscript os-2016-82** by Onken et al.:
"High-Resolution Observations in the Western Mediterranean Sea: The REP14-MED Experiment"

**Action taken on specific points raised by Reviewer #1**
(Page and line numbers refer to the new manuscript)

**Very few results about the calibration and intercalibration of the different sensors (quite exclusively limited to the ship based CTDs).**

An intercomparison of the gliders was never planned: 11 gliders were in use for the oceanographic experiment (plus 5 backup gliders), and they were from 3 different manufacturers (Teledyne, Kongsberg, ALSEAMAR) and operated by 5 different institutions (CMRE, UEA, ALSEAMAR, WTD71, PSU). A real intercomparison would have been too complex in terms of organisational issues; therefore, we relied on the calibration of the manufacturers.

**1. If the paper is the introductory paper for a special number, I would like to see at least some indications on the others papers of the special issue, with some, very rapid mentions of the main results; I guess that authors have already some idea of papers that will be submitted and of their main results.**

The main results of the other papers in the special issue are summarized in the discussion. See P8L22-P11L5.

**2. I cannot understand as, in the present "Argo era", temperature and salinity observations are still not publicly shared (as the authors indicated at the end of the draft) and they are "available only for the partners of the experiment". Although "interested institutions can sign up for partnership at any time", I considered that in the present day ocean science strategy would impose a more open data availability. But it is simply my opinion.**

The  paragraph on *Data availability* has been rewritten. See P12 L14-15.

**Action taken on specific points raised by Reviewer #2**
(Page and line numbers refer to the new manuscript)

**What is the overall scientific goal of the project?**

The overall goal of the REP "projects" is now defined in the Introduction (P2L7-8)

**Scientific questions to be answered?**

The particular objectives of REP14-MED are defined in the Introduction (P2L10-15)

**For what scientific reason were special measurements carried out?**

This is now specified in a new paragraph in Section 4 (P5L5-14)

**What is the scientific goal of the special issue and which questions should be answered?**

The scientific goal of the special issue will be defined in the (upcoming) Editorial.

**What is the scientific content of each single manuscripts of the special issue (describing the results of the measurements)?**

The main results of the other papers in the special issue are summarized in the Discussion. See P8L22-P11L5.

**Additional major changes**
(Page and line numbers refer to the new manuscript; for minor changes see the marked-up manuscript "diff.pdf")

- The paragraph on "general circulation" was rewritten (P2L23-P4L2)
- The paragraph on "CTD casts" was rewritten (P5L5-14)
- The "Conclusions" were rewritten; partly, paragraphs were adopted from the "Discussion" of the old manuscript (P11L6-P12L12)
- "References": more references added and style adjusted to *Ocean Science* rules.
- New Figure 2

In addition, several differences in the captions of tables and figures are highlighted in the marked-up manuscript "diff.pdf"; however, these are not real. Potentially, the programme "latexdiff" got confused because the sequence of tables and figures was switched.

Escudier2016Escudier2016Escudier et al.(2016)  Garcia1996García-Ladona et al.(1996) Hemming2017Hemming et al.(2017) Hernandez2018Hernandez-Lasheras et al.(2018)  Kusel2017Küsel et al.(2017)

[revised manuscript text omitted]